# Multiplier Method for Predicting the Sitting Height Growth at Maturity: A Database Analysis

**DOI:** 10.3390/children9111763

**Published:** 2022-11-17

**Authors:** Julio J. Jauregui, Larysa P. Hlukha, Philip K. McClure, Dror Paley, Mordchai B. Shualy, Maya B. Goldberg, John E. Herzenberg

**Affiliations:** 1Department of Orthopaedics, University of Maryland Medical Center, 110 S. Paca Street, 6th Floor, Suite 300, Baltimore, MD 21201, USA; 2International Center for Limb Lengthening, Rubin Institute for Advanced Orthopedics, Sinai Hospital of Baltimore, 2401 W. Belvedere Ave, Baltimore, MD 21215, USA; 3Paley Orthopedic and Spine Institute, 901 45th St. Kimmel Building, West Palm Beach, FL 33407, USA; 4Beth Tfiloh Dahan Community School, 3300 Old Court Rd, Pikesville, MD 21208, USA

**Keywords:** lumbar spine, skeletal maturity height, spine growth prediction, thoracic spine

## Abstract

This study aims to develop multipliers for the spine and sitting height to predict sitting height at maturity. With the aid of longitudinal and cross-sectional clinical databases, we divided the total sitting height, cervical, thoracic, and lumbar lengths at skeletal maturity by these same four factors at each age for each percentile given. A series of comparisons were then carried out between the multipliers as well as the percentiles and the varied racial and ethnic groups within them. Regarding sitting height, there was little variability and correlated with the multipliers calculated for the thoracic and lumbar spine. The multiplier method has demonstrated accuracy that is not influenced by generation, percentile, race, and ethnicity. This multiplier can be used to anticipate mature sitting height, the heights of the thoracic, cervical, and lumbar spine, as well as the lack of spinal growth after spinal fusion surgery in skeletally immature individuals.

## 1. Introduction

Volumetric growth and ossification of the spine is a remarkably long and slow pro-cess that starts around the third month of prenatal life and continues until the second decade of life. Spine deformity, especially in early life, can negatively affect a growing spine as asymmetric forces act upon the vertebral column’s growth plates. Spinal growth involves more than 130 growth plates that work in synchronicity. Because sitting height is a major component of spinal growth it provides a good parameter to observe and evaluate the progression of spinal growth [1].

Knowledge of spinal growth is of interest to spinal surgeons due to the nature of treating spine-related disorders where the growth ratios between remaining and elapsed growth are important parameters for surgical planning. Paley et al. [2] devised the multi-plier method, a method of limb-length discrepancy and growth prediction. This method uses simple growth coefficients (also known as multipliers) to provide accurate predic-tions of limb length at maturity [3,4]. Spine growth, however, is a dynamic process and does not progress linearly; rather, it consists of periods of acceleration and subsequent de-celeration [1]. Hence, the multipliers do not isolate the growth spurt that occurs during adolescence and the growth that remains is a deciding factor for the intensifying of spinal deformities such as congenital scoliosis, kyphoscoliosis, and angular thoracolumbar ky-phosis in patients with achondroplasia.

Sitting height consists of pelvic height, spine height, and skull height, and therefore serves as a capable tool for predicting spinal growth at maturity [5,6,7]. The known pub-lished data assessing spinal growth based on sitting-height measurements either focuses mostly on one social class or ethnic group or does not stratify by age or sex. We intended to consolidate the databases that assess sitting height and develop multipliers that predict sitting height to aid in spine deformity correction.

## 2. Materials and Methods

We calculated and compared the multipliers for different percentiles of 16 databases with sitting-height measurement data [1,5,6,7,8,9,10,11,12,13,14,15,16,17,18]. Fredriks et al. provided the most compre-hensive assessment of spinal growth for each percentile group at each age [9]. At each age and percentile group, we divided the sitting height at skeletal maturity (Lm) by the corre-sponding sitting height at each given age (L): (M = Lm/L). We compared these multipliers (M) for each percentile group at each age. In addition, we calculated multipliers from two radiographic databases that contained measurements of the lumbar, thoracic, and cervical spine segments. These were largely obtained from standing or sitting radiographs. The databases provided measurements for a range of ages and included measurements ob-tained until subjects achieved skeletal maturity.

All data were tabulated using spreadsheet software (Excel; Microsoft, Redmond, WA, USA) with the aid of a statistical program (SPSS; IBM, Armonk, NY, USA). A series of polynomial regression analyses were used to evaluate the correlation between mean values in each of the databases using Fredriks et al.’s data as the “gold standard” database due to its inclusion of important variables such as age, ethnicity, and congenital variations. The mean sitting height that was reported for each age and stratified by sex in each of the databases was compared with the mean sitting height and the previously described SDs.

## 3. Results

We found that in Fredriks et al.’s database [9], the multipliers for each SD (−2.5 SD, −1 SD, 0 SD, +1 SD, +2.5 SD) at each given age for both males and females (mean variability ±0.065; maximum variability ±0.372) were similar. We also found that the variability between the multipliers was highest at birth, and generally decreased as age increased (Figure 1 and Figure 2).

We then compared the mean multipliers of sitting height to the multipliers for the cervical, thoracic, and lumbar segments of the spine. The multipliers of the thoracic and lumbar segments were nearly identical to those of sitting height (mean variability ±0.012; max variability ±0.043). The cervical multipliers, however, were substantially higher than those calculated from sitting height. The cervical multipliers are virtually identical to upper-extremity multipliers (mean variability ±0.0489; max variability ±0.060) [2]. See Figure 3 for a detailed comparison of all multipliers.

To diversify our data and ensure accuracy, we incorporated sitting heights from various continents and countries into our research. Zivicnjak et al. [13] published a cross-sectional study documenting the growth of 5155 (2591 female and 2564 male) Croatian children. The subjects were selected from various schools and kindergartens within the town of Zagreb to represent the entire socioeconomic spectrum. The multipliers calculated from Zivicnjak et al.’s data were virtually identical to those calculated by Fredriks et al. [9]. Pathmanathan and Prakash [14] also conducted a cross-sectional survey of northwestern Indian children. They measured the sitting height, leg length, and weight in 668 children (327 females and 341 males) between ages 6 and 16. Of the 668 children, 80% were measured more than once yearly. The multipliers calculated from this study correlated closely with those of Fredriks et al.

Our predicted multiplier method requires only a single measurement of sitting height and a simple arithmetic operation (Figure 4) to predict sitting height at skeletal maturity (Table 1, Table 2 and Table 3).

## 4. Discussion

The prediction of adolescent sitting height is germane to pediatricians and spinal surgeons. Other methods of growth prediction require both the determination of the sub-ject’s growth percentile and the use of a graph, all of which we have now consolidated in-to one simple formula (Figure 4).

Aldegheri and Agostini [19] produced a study compiled from previously published information on sitting height, but the ethnicity or race of subjects is not included. Only the means of sitting height are included without any SDs. Although the female multipliers generated show little variation from those generated from Frederik et al.’s data, the devia-tion between the male multipliers is considerably greater. While we are unsure of the exact reasons for the disparity between the Aldegheri and Fredriks et al. studies until age seven, we believe it has to do with the small sample size.

This work has limitations. Even though the sitting-height multiplier is meant to be universal, and we found consistency throughout all studied populations after comparing growth in many different studies, we believe that ultimately the physician should deter-mine if each patient is following the growth curves presented by the multiplier. We also believe that whenever assessing spinal growth alterations and pathologies requiring spi-nal fusion or epiphysiodesis, the spinal growth will not necessarily follow the expected growth. Spinal growth can involve vertebrae changes as well as changes to the vertebral disk height. This may have implications for changes in the sitting height after maturity [18].

Longitudinal studies have demonstrated growth patterns more accurately than cross-sectional studies, as they have traced the growth of individuals over an elapsed pe-riod. However, there are far fewer longitudinal studies available due to the high cost and time limitations to perform such studies. A territory-wide cross-sectional study by Leung et al. [10] evaluated the sitting height of 25,000 Chinese children of higher socioeconomic status from birth through 18 years of age. Similarly, Lin et al. [11] assessed the sitting heights of Han children (the main ethnic group in China) and 27 minority groups. In total, 409,941 Han children and 70,298 minority children between the ages of 7 and 18 were measured. The authors reported that the sitting heights recorded from the Han cohort were generally higher than those recorded from the minority groups. However, when comparing the multipliers of Leung et al. [10] and both groups of Lin et al. [11] to the mul-tipliers generated from Fredriks et al. [9] data, a minimal difference is observed.

Lee et al. [12] published a longitudinal study observing 1,139 healthy Taipei school-children ranging from 8 to 18 years of age. In addition to sitting height, they also docu-mented arm span, skinfold thickness, and body mass index from 1994 through 1997. Alt-hough it has been suggested that longitudinal studies hold more accuracy than cross-sectional, the Lee et al. [12] multipliers held little variability to those of Fredriks et al. [9].

The multiplier method also can determine the appropriate timing of arthrodesis, as well as to predict the amount of height lost because of a specific spine operation. Each vertebra in the lumbar and thoracic spine make approximately equal growth contributions—i.e., any single vertebra in the lumbar spine contributes approximately one-fifth of the total growth of the lumbar spine, and any single vertebra in the thoracic spine contrib-utes approximately one-twelfth of the total growth of the thoracic spine [15,20]. Therefore, to predict the height of a single lumbar vertebra at maturity (Llv), multiply the current length of the lumbar spine (Lls) by the multiplier for the current age (M) and divide the result by five ([LlsM]/5 = Llv). Similarly, to predict the height of a single thoracic vertebra at maturity (Ltv), multiply the current length of the lumbar spine (Lts) by the multiplier for the current age (M) and divide the result by twelve ((LtsM)/12 = Ltv) (Figure 5).

To calculate the growth lost as the result of arthrodesis, one must first calculate the growth remaining in the affected area of the spine. Growth remaining (G) is equal to the length of the relevant section of the spine at maturity minus the current length of the rele-vant section (M[L-1] = G). Thus, to calculate the growth lost as a result of arthrodesis (Ge), multiply the growth remaining for a single vertebra in the relevant segment of the spine ([(Lls(M-1)]/5 for lumbar vertebrae); ([Lts(M-1)]/12 for thoracic vertebrae) by the number of vertebrae fused (Vf) (Figure 5).

The prediction of sitting height is an important factor in the planning of limb length-ening for achondroplastic dwarfism [3]. As demonstrated by Paley et al., there is a close correlation between the sitting heights of these patients and normal subjects [3,16,20,21,22]. Hence, our sitting height multiplier also can be employed for achondroplastic patients [3]. In normal adults, sitting height comprises 53% of the total standing height at skeletal ma-turity [23,24,25]. Therefore, when lower extremity lengthening is planned in achondroplastic patients, the goal is to obtain a length in the lower extremity of 88.6% (47/53) of the pre-dicted sitting height, to achieve Vitruvian proportions.

Paley et al. have shown that the multipliers are independent of percentile groupings, height, generation, socioeconomic class, ethnicity, and race, and this independence has been clinically validated [2,4,17]. Such a relationship could be considered a mathematical property of human growth and development. The relative size and curve of the various multipliers are indicators of normal proportion; the lower extremity, having the highest multiplier, increases the most proportionally, from 35% of the height at birth to 47% of the height at skeletal maturity [20]. Sitting height, on the other hand, has a lower multiplier, decreasing proportionally from 65% of the height at birth to 53% at skeletal maturity.

## 5. Conclusions

Most known current data that has attempted to accurately predict spinal growth at maturity focuses on either specific ethnic groups or concentrates on a particular patient population—such those with congenital scoliosis or achondroplasia. We evaluated available databases on sitting height and multipliers for spinal growth and consolidated them to develop a multiplier for predicting sitting height growth at maturity. This method can also aid in calculating growth modification resulting from arthrodesis as well as facilitaing planning for limb lengthening procedures with challenging diagnoses such as achondroplasic dwarfism.

## Figures and Tables

**Figure 1 children-09-01763-f001:**
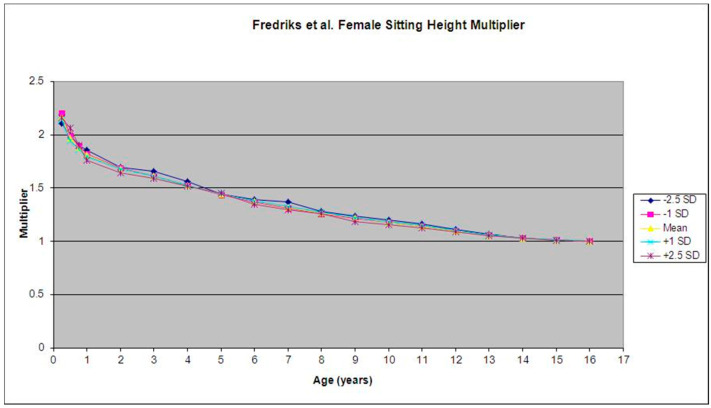
Mean and standard deviation for the female sitting height multiplier were calculated using data from Fredriks et al. [9].

**Figure 2 children-09-01763-f002:**
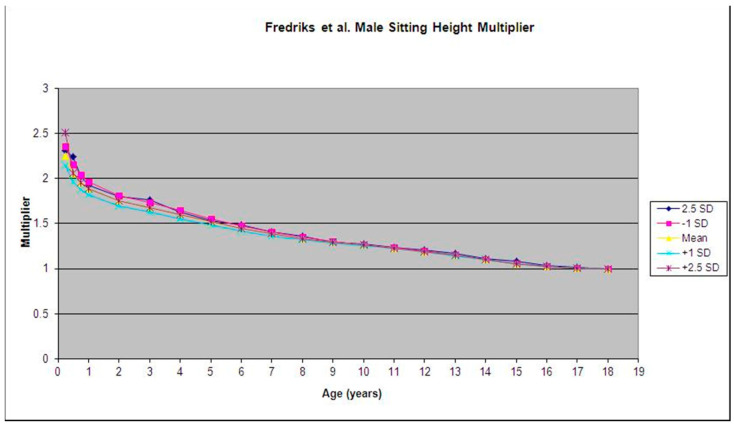
Mean and standard deviation for the male sitting height multiplier were calculated using data from Fredriks et al. [9].

**Figure 3 children-09-01763-f003:**
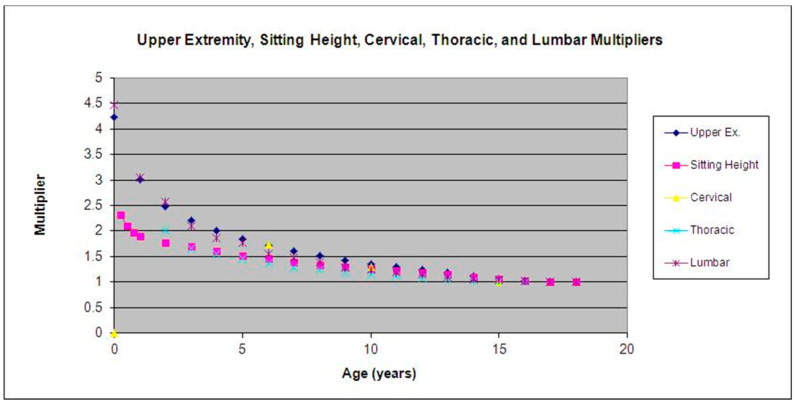
Upper extremity, sitting height, cervical, thoracic, and lumbar multipliers.

**Figure 4 children-09-01763-f004:**
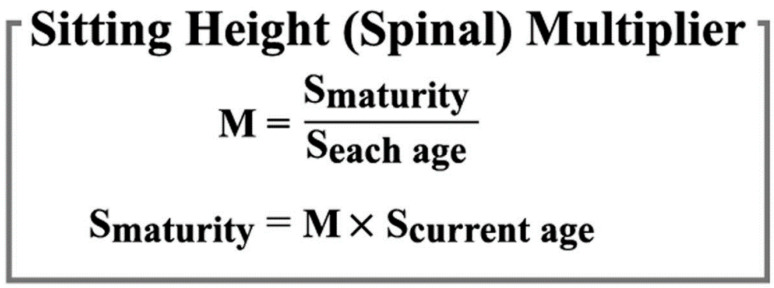
Equation for spinal sitting height multiplier. M, age-specific multiplier; S, age-specific sitting height.

**Figure 5 children-09-01763-f005:**
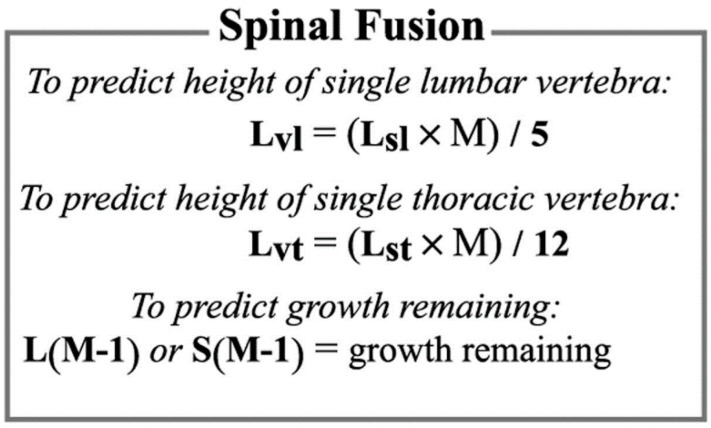
Equation for predicting height of the single lumbar and thoracic vertebrae and remaining growth. L, age-specific length; Lsl, age-specific length of lumbar spine; Lst, age-specific length of thoracic spine; Lvl, length of single lumbar vertebra at maturity; Lvt, length of single thoracic vertebra at maturity; M, age-specific multiplier; S, age-specific sitting height.

**Table 1 children-09-01763-t001:** Sitting height multiplier for 0 to 3 years of age.

Age (Years + Months)	Boys	Girls	Age (Years + Months)	Boys	Girls
0 + 1	-	-	3 + 1	1.688	1.611
0 + 2	-	-	3 + 2	1.681	1.603
0 + 3	2.315	2.153	3 + 3	1.673	1.596
0 + 4	2.165	2.052	3 + 4	1.666	1.589
0 + 5	2.131	2.024	3 + 5	1.658	1.581
0 + 6	2.096	1.995	3 + 6	1.651	1.574
0 + 7	2.04	1.944	3 + 7	1.643	1.567
0 + 8	2.006	1.916	3 + 8	1.635	1.559
0 + 9	1.971	1.887	3 + 9	1.628	1.552
0 + 10	1.968	1.839	3 + 10	1.62	1.545
0 + 11	1.934	1.839	3 + 11	1.613	1.537
**1**	**1.899**	**1.81**	**4**	**1.605**	**1.53**
1 + 1	1.888	1.799	4 + 1	1.598	1.523
1 + 2	1.877	1.788	4 + 2	1.591	1.516
1 + 3	1.865	1.777	4 + 3	1.585	1.508
1 + 4	1.854	1.765	4 + 4	1.578	1.501
1 + 5	1.843	1.754	4 + 5	1.571	1.494
1 + 6	1.832	1.743	4 + 6	1.564	1.487
1 + 7	1.82	1.732	4 + 7	1.557	1.479
1 + 8	1.809	1.721	4 + 8	1.55	1.472
1 + 9	1.798	1.71	4 + 9	1.544	1.465
1 + 10	1.787	1.698	4 + 10	1.537	1.458
1 + 11	1.775	1.687	4 + 11	1.53	1.45
**2**	**1.764**	**1.676**	**5**	**1.523**	**1.443**
2 + 1	1.758	1.671	5 + 1	1.517	1.437
2 + 2	1.753	1.666	5 + 2	1.512	1.431
2 + 3	1.747	1.662	5 + 3	1.506	1.425
2 + 4	1.741	1.657	5 + 4	1.5	1.419
2 + 5	1.736	1.652	5 + 5	1.494	1.413
2 + 6	1.73	1.647	5 + 6	1.489	1.408
2 + 7	1.724	1.642	5 + 7	1.483	1.402
2 + 8	1.719	1.637	5 + 8	1.477	1.396
2 + 9	1.713	1.633	5 + 9	1.471	1.39
2 + 10	1.707	1.628	5 + 10	1.466	1.384
2 + 11	1.702	1.623	5 + 11	1.46	1.378
**3**	**1.696**	**1.618**	**6**	**1.454**	**1.372**

**Table 2 children-09-01763-t002:** Sitting height multiplier for 6+ to 9 years of age.

Age (Years + Months)	Boys	Girls	Age (Years + Months)	Boys	Girls
6 + 1	1.449	1.368	9 + 1	1.291	1.212
6 + 2	1.443	1.364	9 + 2	1.288	1.209
6 + 3	1.438	1.36	9 + 3	1.286	1.206
6 + 4	1.432	1.356	9 + 4	1.283	1.203
6 + 5	1.427	1.352	9 + 5	1.281	1.2
6 + 6	1.422	1.348	9 + 6	1.278	1.197
6 + 7	1.416	1.344	9 + 7	1.276	1.193
6 + 8	1.411	1.34	9 + 8	1.273	1.19
6 + 9	1.405	1.336	9 + 9	1.271	1.187
6 + 10	1.4	1.332	9 + 10	1.268	1.184
6 + 11	1.394	1.328	9 + 11	1.266	1.181
**7**	**1.389**	**1.324**	**10**	**1.263**	**1.178**
7 + 1	1.385	1.319	10 + 1	1.26	1.175
7 + 2	1.381	1.315	10 + 2	1.258	1.172
7 + 3	1.376	1.31	10 + 3	1.255	1.169
7 + 4	1.372	1.305	10 + 4	1.252	1.166
7 + 5	1.368	1.3	10 + 5	1.249	1.163
7 + 6	1.364	1.296	10 + 6	1.247	1.161
7 + 7	1.359	1.291	10 + 7	1.244	1.158
7 + 8	1.355	1.286	10 + 8	1.241	1.155
7 + 9	1.351	1.281	10 + 9	1.238	1.152
7 + 10	1.347	1.277	10 + 10	1.236	1.149
7 + 11	1.342	1.272	10 + 11	1.233	1.146
**8**	**1.338**	**1.267**	**11**	**1.23**	**1.143**
8 + 1	1.334	1.263	11 + 1	1.227	1.14
8 + 2	1.331	1.258	11 + 2	1.224	1.136
8 + 3	1.327	1.254	11 + 3	1.221	1.133
8 + 4	1.323	1.25	11 + 4	1.217	1.129
8 + 5	1.319	1.245	11 + 5	1.214	1.126
8 + 6	1.316	1.241	11 + 6	1.211	1.122
8 + 7	1.312	1.237	11 + 7	1.208	1.119
8 + 8	1.308	1.232	11 + 8	1.205	1.115
8 + 9	1.304	1.228	11 + 9	1.202	1.112
8 + 10	1.301	1.224	11 + 10	1.198	1.108
8 + 11	1.297	1.219	11 + 11	1.195	1.105
**9**	**1.293**	**1.215**	**12**	**1.192**	**1.101**

**Table 3 children-09-01763-t003:** Sitting height Multiplier for 12+ to 15 years of age.

Age (Years + Months)	Boys	Girls	Age (Years + Months)	Boys	Girls
12 + 1	1.189	1.098	15 + 1	1.056	1.011
12 + 2	1.185	1.094	15 + 2	1.054	1.01
12 + 3	1.182	1.091	15 + 3	1.051	1.009
12 + 4	1.178	1.088	15 + 4	1.048	1.008
12 + 5	1.175	1.084	15 + 5	1.045	1.007
12 + 6	1.171	1.081	15 + 6	1.043	1.006
12 + 7	1.168	1.078	15 + 7	1.04	1.005
12 + 8	1.164	1.074	15 + 8	1.037	1.004
12 + 9	1.161	1.071	15 + 9	1.034	1.003
12 + 10	1.157	1.068	15 + 10	1.032	1.002
12 + 11	1.154	1.064	15 + 11	1.029	1.001
**13**	**1.15**	**1.061**	**16**	**1.026**	**1**
13 + 1	1.146	1.058	16 + 1	1.025	
13 + 2	1.142	1.056	16 + 2	1.023	
13 + 3	1.138	1.053	16 + 3	1.022	
13 + 4	1.133	1.051	16 + 4	1.021	
13 + 5	1.129	1.048	16 + 5	1.019	
13 + 6	1.125	1.046	16 + 6	1.018	
13 + 7	1.121	1.043	16 + 7	1.017	
13 + 8	1.117	1.04	16 + 8	1.015	
13 + 9	1.113	1.038	16 + 9	1.014	
13 + 10	1.108	1.035	16 + 10	1.013	
13 + 11	1.104	1.033	16 + 11	1.011	
**14**	**1.1**	**1.03**	**17**	**1.01**	
14 + 1	1.097	1.029	17 + 1	1.009	
14 + 2	1.093	1.027	17 + 2	1.008	
14 + 3	1.09	1.026	17 + 3	1.008	
14 + 4	1.086	1.024	17 + 4	1.007	
14 + 5	1.083	1.023	17 + 5	1.006	
14 + 6	1.08	1.021	17 + 6	1.005	
14 + 7	1.076	1.02	17 + 7	1.004	
14 + 8	1.073	1.018	17 + 8	1.003	
14 + 9	1.069	1.017	17 + 9	1.003	
14 + 10	1.066	1.015	17 + 10	1.002	
14 + 11	1.062	1.014	17 + 11	1.001	
**15**	**1.059**	**1.012**	**18**	**1**	

## Data Availability

The data presented in this study are available upon request from the corresponding author.

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
