# Peer review of "Multiplier Method for Predicting the Sitting Height Growth at Maturity: A Database Analysis"

_children, 2022, doi:10.3390/children9111763_

Round 1
Reviewer 1 Report
Overview
The authors present an interesting manuscript that describes modifiers which can be used to predict the sitting height and spinal height at skeletal maturity, using large databases of skeletally immature individuals. The manuscript is well written and the introduction and discussion were clear. However, I have recommendations for the authors to add more details in several places, and move some of the text into different sections.
Major comments
1. Could you add a descriptor of the study design in the title? E.g., cross-sectional, longitudinal, or perhaps more succinctly “Database Analysis”. Methodological term(s) in the title will help readers understand that this is an original research paper, rather than a review paper, cohort study, case series, or other design. Also consider adding “at maturity” after “growth”, or you could incorporate a term about youth such as “skeletally immature”.
2. I think you could make your objectives/aims/background a little clearer. To my understanding, it seems like you are validating or building on a previously-identified calculation from the Fredrik et al study using a broader population sample, or perhaps you are replicating their study using a larger or diverse sample of databases. Regardless of if my understanding is correct, the exact relationship between your work and the Fredrik study (or other previous studies) should be a little clearer, and I think you should devote a couple sentences to this in the Introduction and also make your aims clearer to point out what is new and what has already been done before, and if you are aiming to build on previous research. In part, I was confused why your paper says you will “develop modifiers” if these have already been developed previously, even in a limited manner. It’s not clear what part of your study represents new material.
3. Much of the information in the Materials and Methods section that talks about previous research on this topic should be included in the Introduction. This will introduce the prior studies on the topic and better prepare readers to understand the Methods. See the information in this section starting with “Many proposed…” including the following paragraph. Readers should have a good understanding of the rationale for why you did the study before getting to the Methods section.
4. Can you better describe the databases used in your study? I think a table describing the size (number of participants), age range, mean or median age, and any other pertinent info for each database such as country, sex % (male/female) would be highly relevant considering your conclusions talk about a broad generalizability. I was confused because there were at least 15 databases studied, but it’s not clear what is in these databases or how they can be looked up by the reader. You reference some of them in the text, but the number referenced does not seem to add up to 15. Consider adding any information about the databases and age ranges or number of patients into the abstract as well in a very concise way. I was also a little confused about how the radiographs were taken in the databases, and if they were all done in a standard way like standing, lying down, sitting, etc. If this varies it’s OK but it would be nice to mention.
5. Ethics - Are all of the databases you used referenced? Is the data publicly available? Part of this will help inform the readers’ interpretation of the ethical approval not being required. There is a contrast between the ethics statement and the information in the methods that you could help clarify. You write “The data are not publicly available due to privacy concerns with protected health information”- however, I’m confused because if there is protected health information in non-public databases in this study, wouldn’t ethical approval be needed for you to do research with it? Some IRBs grant a waiver for de-identified databases or data that is already collected (like a literature review). Perhaps you are doing only a secondary analysis of data?
6. I was confused why some of the Results are shown in the Discussion section. The tables 1 and 2 with the “Sitting height multiplier” values appear to be results – can you move them in the Results section?
7. You did a multiple linear regression – I had a hard time finding the full results for the regression models. Is there a measure of precision, such as confidence intervals, p-value, or a coefficient for your regression model(s)? If it’s lengthy to include, you could consider posting it as a supplemental file for those interested in seeing the full results, however it would greatly strengthen your manuscript to include some of these details along with the results.
Minor comments
1. The phrase "as well as the amount of height lost due to spinal fusion surgery” could be altered in the abstract. Is height of the spine truly lost or is it simply not gained after surgery? (e.g., a lack of spinal growth after spinal fusion surgery in skeletally immature individuals). Is there a better way to word this?
2. The Keywords below the abstract - “spine multipliers” and “height at maturity” for this article are not well-recognized terms in my opinion. I could not find them in the standard place – please look here: https://meshb.nlm.nih.gov/. Consider adding different terms, perhaps more broad terms in the subject matter, such as “growth charts” or “spinal fusion” or “body height”
3. Statistical analysis – “using Fredriks et al’s data as the independent variable” – can you be explicit about which data item was the independent variable? Also, can you list the dependent variable(s) / covariate(s) used in the model(s)?
Comments for clarity
1. Abstract aim – at the end of the 1st sentence, should “height” be “sitting height” ? You are not predicting the standing height so it would be helpful to be specific here.
2. Abstract – “The cervical spine multipliers were nearly identical to the multipliers for upper extremities.” – is this necessary to include in the abstract? Upper extremity data seemed to be a secondary objective or discussion point, so I think you could omit it here.
3. Abstract – “The multipliers of different national and racial groups were also the same” – is it accurate to say they were the same? (this seems to imply they were exactly the same) Maybe they were not significantly different? Or maybe they were similar?
4. Abstract – there is some redundancy between the 3nd to last and the last sentences. E.g., “predicting…” “can be used to predict” – these sentences seem to overlap quite a bit. I think only 1 sentence would be needed to convey this concept.
5. Figure 3 – is there a reason why the dots are not connected with a line like in the other graphs? Also, is there a reason why there is a yellow triangle marker at 0,0 – is that accurate? Last – the caption says there is a standard deviation shown but I don’t see error bars.
6. The term “significant” or “significantly” appears a few times but I did not see any statistical measure of significance or precision
Author Response
Reviewer #1 comments
Comments and Suggestions for Authors
Overview
The authors present an interesting manuscript that describes modifiers which can be used to predict the sitting height and spinal height at skeletal maturity, using large databases of skeletally immature individuals. The manuscript is well written and the introduction and discussion were clear. However, I have recommendations for the authors to add more details in several places, and move some of the text into different sections.
Major comments
- Could you add a descriptor of the study design in the title? E.g., cross-sectional, longitudinal, or perhaps more succinctly “Database Analysis”. Methodological term(s) in the title will help readers understand that this is an original research paper, rather than a review paper, cohort study, case series, or other design. Also consider adding “at maturity” after “growth”, or you could incorporate a term about youth such as “skeletally immature”.
Author response: Thank you for the comments. The title has been modified to reflect your suggestion.
I think you could make your objectives/aims/background a little clearer. To my understanding, it seems like you are validating or building on a previously-identified calculation from the Fredrik et al study using a broader population sample, or perhaps you are replicating their study using a larger or diverse sample of databases. Regardless of if my understanding is correct, the exact relationship between your work and the Fredrik study (or other previous studies) should be a little clearer, and I think you should devote a couple sentences to this in the Introduction and also make your aims clearer to point out what is new and what has already been done before, and if you are aiming to build on previous research. In part, I was confused why your paper says you will “develop modifiers” if these have already been developed previously, even in a limited manner. It’s not clear what part of your study represents new material.
Author response: Thank you for the suggestions. We modified the text throughout the introduction and methods sections to clarify the objectives of the paper as well as methodology utilized to develop the multiplier method for sitting height.
- Much of the information in the Materials and Methods section that talks about previous research on this topic should be included in the Introduction. This will introduce the prior studies on the topic and better prepare readers to understand the Methods. See the information in this section starting with “Many proposed…” including the following paragraph. Readers should have a good understanding of the rationale for why you did the study before getting to the Methods section.
Author response: Thank you, we have modified the first paragraph of the Material and Methods section and moved some text components to the introduction section. We have also removed the subsection headers for the remaining Material & Methods text and have opted to present it as one section split into paragraphs.
- Can you better describe the databases used in your study? I think a table describing the size (number of participants), age range, mean or median age, and any other pertinent info for each database such as country, sex % (male/female) would be highly relevant considering your conclusions talk about a broad generalizability. I was confused because there were at least 15 databases studied, but it’s not clear what is in these databases or how they can be looked up by the reader. You reference some of them in the text, but the number referenced does not seem to add up to 15. Consider adding any information about the databases and age ranges or number of patients into the abstract as well in a very concise way. I was also a little confused about how the radiographs were taken in the databases, and if they were all done in a standard way like standing, lying down, sitting, etc. If this varies it’s OK but it would be nice to mention.
Author response: Thank you for this comment. We made the sitting height databases used for this study more clear in line 55 of the Material and Methods section. Unfortunately, reproducing the exact tables and figures from pre-existing studies without permission of the journals in which they were published would not be possible. We do, however, extensively discuss relevant parts of their data in the Discussion section. We agree mentioning the methods for obtaining radiographs can be helpful. After reviewing the databases, it seems many have used the standard methods (standing/sitting), although some do not explicitly mention this or any variability. Some databases do not address the method of taking radiographs and others specifically concentrate on measuring the radiograph rather than the method use to obtain them.
- Ethics - Are all of the databases you used referenced? Is the data publicly available? Part of this will help inform the readers’ interpretation of the ethical approval not being required. There is a contrast between the ethics statement and the information in the methods that you could help clarify. You write “The data are not publicly available due to privacy concerns with protected health information”- however, I’m confused because if there is protected health information in non-public databases in this study, wouldn’t ethical approval be needed for you to do research with it? Some IRBs grant a waiver for de-identified databases or data that is already collected (like a literature review). Perhaps you are doing only a secondary analysis of data?
Author response: Thank you, the statement about privacy concerns was a mistake. The data researched is publicly available. This has been changed at line 212 to read “The data presented in this study are available upon request from the corresponding author.”
- I was confused why some of the Results are shown in the Discussion section. The tables 1 and 2 with the “Sitting height multiplier” values appear to be results – can you move them in the Results section?
Author response: We agree. The tables describing sitting-height multipliers for each age range have been moved to the Results section. Their callout is now at line 104 and the figure and tables follow immediately after.
- You did a multiple linear regression – I had a hard time finding the full results for the regression models. Is there a measure of precision, such as confidence intervals, p-value, or a coefficient for your regression model(s)? If it’s lengthy to include, you could consider posting it as a supplemental file for those interested in seeing the full results, however it would greatly strengthen your manuscript to include some of these details along with the results.
Author response: Thank you for the comment. We can see how this may be of interest to a particular reader. We have not been able to locate, prepare, and format the relevant data in a timely manner while trying to meet the deadline of this revision request, but we hope that you agree that incorporating this now is nonessential, as any interested party can contact the corresponding author directly in the future as per the data availability statement and we will be able to have the time to fulfill such requests.
Minor comments
- The phrase "as well as the amount of height lost due to spinal fusion surgery” could be altered in the abstract. Is height of the spine truly lost or is it simply not gained after surgery? (e.g., a lack of spinal growth after spinal fusion surgery in skeletally immature individuals). Is there a better way to word this?
Author response: Thank you for the comment. We modified the Abstract for clarification of this phrasing at lines 24-25.
The Keywords below the abstract - “spine multipliers” and “height at maturity” for this article are not well-recognized terms in my opinion. I could not find them in the standard place – please look here: https://meshb.nlm.nih.gov/. Consider adding different terms, perhaps more broad terms in the subject matter, such as “growth charts” or “spinal fusion” or “body height”
Author response: Thank you. We have modified this to reflect readily recognizable terms such as skeletal maturity height, lumbar spine, spine growth prediction, thoracic spine at line 26
- Statistical analysis – “using Fredriks et al’s data as the independent variable” – can you be explicit about which data item was the independent variable? Also, can you list the dependent variable(s) / covariate(s) used in the model(s)?
Author response: Independent variable may not be the correct phrasing for the analysis performed. Since Fredriks et al. presented the most comprehensive data on sitting-height measurement (including ethnicity, age, congenital alterations etc.) we used it as the means of a gold standard when comparing these variables from other databases. We rephrased this sentence to reflect the analysis as such, at line 67.
Comments for clarity
- Abstract aim – at the end of the 1stsentence, should “height” be “sitting height” ? You are not predicting the standing height so it would be helpful to be specific here.
Author response: Yes. Thank you. Updated at line 17.
- Abstract – “The cervical spine multipliers were nearly identical to the multipliers for upper extremities.” – is this necessary to include in the abstract? Upper extremity data seemed to be a secondary objective or discussion point, so I think you could omit it here.
Author response: Thank you. The sentence has been omitted.
- Abstract – “The multipliers of different national and racial groups were also the same” – is it accurate to say they were the same? (this seems to imply they were exactly the same) Maybe they were not significantly different? Or maybe they were similar?
Author response: Thank you. We modified the phrasing in the Abstract and removed that phrase.
- Abstract – there is some redundancy between the 3ndto last and the last sentences. E.g., “predicting…” “can be used to predict” – these sentences seem to overlap quite a bit. I think only 1 sentence would be needed to convey this concept.
Author response: Thank you for the comment. We rephrased the Abstract to rid of some redundancy.
- Figure 3 – is there a reason why the dots are not connected with a line like in the other graphs? Also, is there a reason why there is a yellow triangle marker at 0,0 – is that accurate? Last – the caption says there is a standard deviation shown but I don’t see error bars.
Author response: Thank you for the comment. Graphs 1 and 2 refer to calculations based on the databases to attempt standardization of sitting height at different ages. These data points are connected to further clarify the general downward trend with increased age. Since Figures 1 and 2 showed some variability we believed that including the mean and SD was pertinent. Figure 3 compared sitting height multipliers to those of spinal segments. Since negligible variability was found in all but cervical spine multipliers, we only included the means and SD data in the text rather than figures (the original figure name was mislabeled). Also no line is present in Figure 3 because this was a comparison of multiplier methods and not a delineation of a trend. The yellow triangle is part of the data point for cervical spine multiplier.
- The term “significant” or “significantly” appears a few times but I did not see any statistical measure of significance or precision
Author response: Thank you for the suggestion. We replace the word significant where no statistical data is assessed with more appropriate synonyms (lines 87 and 191).
Reviewer 2 Report
Thank you for the opportunity to review this interesting article. Please, I encourage to the authors to answer me about the following concerns:
In abstract section.
-Please add one or two sentences about background before the objective.
In introduction section.
-Overall, to include only 8 references is very poor. I recommed to the authors to increase the number of references.
-Please, add examples of spine deformity that can negatively affect the growing spine. It can facilitate the comprehension.
-Please add the year in "Paley et al (xxxx)".
-Authors use the reference number 1 a lot. It is not recommedable.
-Please, incorporate information about previous studies with the same objective that yours.
In methods section.
-Please, explain better your multiplier.
In results.
-I recommed that authors study how the results section must be performed. I have not clear if they use patients or registers previously published.
In discussion.
-Please, unify the size of letter. I do not understand these mistakes when the template is provided for the Journal.
-Where are the limitations subsection?
This article do not present conclusion. This is an essential part of an article.
Please, revise the english version.
Author Response
Reviewer #2 comments
Comments and Suggestions for Authors
Thank you for the opportunity to review this interesting article. Please, I encourage to the authors to answer me about the following concerns:
In abstract section.
-Please add one or two sentences about background before the objective.
Author response: Thank you for the comment. We added the justification for use of sitting height as a reliable measure to determine spinal growth at maturity as well as explanation for use of multipliers to determine the sitting height in lines 41-52 in the Introduction section.
In introduction section.
-Overall, to include only 8 references is very poor. I recommed to the authors to increase the number of references.
Author response: Thank you for the comment. We use an overall of 29 references in the manuscript. We have added additional references in the Introduction paragraph pertaining to description of sitting-height in line 48.
-Please, add examples of spine deformity that can negatively affect the growing spine. It can facilitate the comprehension.
Author response: Thank you for the comment. We included some examples of spinal deformity which can negative effect the growing spine in the Introduction section in lines 44-46.
-Please add the year in "Paley et al (xxxx)".
Author response: We have included the bracketed reference citation per journal style. Although multiple references have Paley as first author, the bracketed references prevent ambiguity as to which is which. The manuscript template for Children (Basel) states “References should be numbered in order of appearance and indicated by a numeral or numerals in square brackets—e.g., [1] or [2,3], or [4–6].”
-Authors use the reference number 1 a lot. It is not recommedable.
Author response: Thank you for the comment. We use reference 1 three times in the manuscript two of which are in the Introduction paragraph discussing spinal growth. The third is included in with the other 15 databases referenced for sitting height data.
-Please, incorporate information about previous studies with the same objective that yours.
Author response: Thank you for your comment. We discuss Fredrich et al, which provides the most comprehensive data on sitting-height which we used as a gold standard to develop sitting-height multipliers. We also discuss Paley et al. which developed multipliers for predicting limb length in upper and lower extremities (separate studies). We also extensively discuss multiple other studies either assessing sitting-height data or developing multipliers for other parameters different then the spine. No other study currently exists with the same objective we set out to predict in our manuscript which is sitting-hight multipliers for the growing spine at maturity.
In methods section.
-Please, explain better your multiplier.
Author response: Thank you for the comment. We modified the Materials and Methods section to clarify where the data for sitting height was calculated from, how we compared the percentile groups, and our statistical analysis parameters. Please see lines 58-61 and 65-70.
In results.
-I recommed that authors study how the results section must be performed. I have not clear if they use patients or registers previously published.
Author response: Thank you for the comment. We modified the Results section to include our main published reference database (line 85) as well as figures and tables we calculated from the referenced database (line 104).
In discussion.
-Please, unify the size of letter. I do not understand these mistakes when the template is provided for the Journal.
Author response: The size of the text within the body of the manuscript and its tables is uniform and adherent to the journal template. We have provided two figures (figure #s 4 and 5) which are images of formulas. We believe these are what the reviewer is referencing with this comment. As these are technically figures and not manuscript text, we believe this is acceptable. The journal’s production team may manipulate the dimensions of the figures as they see fit, of course.
-Where are the limitations subsection?
Author response: A paragraph exists that discusses limitations. This can be found in lines 127-135. The manuscript template for Children (Basel) does not require a specific subsection for limitations.
This article do not present conclusion. This is an essential part of an article.
Author response: Thank you for the comment. We added a Conclusion section in lines 194-202. The manuscript template for Children (Basel) states “Conclusions: This section is not mandatory but can be added to the manuscript if the discussion is unusually long or complex.”
Round 2
Reviewer 1 Report
I have no comments for review and appreciate the revisions that you made on the manuscript.
Reviewer 2 Report
Authors have provided answer to my concerns.